# Enhancing Paper Packaging’s Wet Strength Using the Synergy between Chitosan and Nanofibrillated Cellulose Additives

**DOI:** 10.3390/polym16020227

**Published:** 2024-01-12

**Authors:** Laura Andze, Marite Skute, Juris Zoldners, Martins Andzs, Gatis Sirmulis, Ilze Irbe, Ulla Milbreta, Inga Dabolina, Inese Filipova

**Affiliations:** 1Latvian State Institute of Wood Chemistry, Dzerbenes Street 27, LV-1006 Riga, Latvia; polarlapsa@inbox.lv (M.S.); jzoldn@inbox.lv (J.Z.); martins.andzs@gmail.com (M.A.); ilze.irbe@kki.lv (I.I.); ulla.milbreta@gmail.com (U.M.); inese.filipova@kki.lv (I.F.); 2Ltd. V.L.T., Murmuizas Street 11a, LV-4201 Valmiera, Latvia; tehnologs@eggbox.lv; 3Faculty of Natural Science and Technology, Riga Technical University, 6A Kipsalas Street, LV-1048 Riga, Latvia; 4Personal Protective Equipment Laboratory, Riga Technical University, Kipsalas 6B-242, LV-1048 Riga, Latvia; inga.dabolina@rtu.lv

**Keywords:** chitosan, fibrillated nanocellulose, wet strength, tensile index, Cobb, stretch, hemp fibers, kraft fibers, recovered paper fibers, paper properties

## Abstract

The demand for eco-friendly packaging materials has urged researchers to look for alternatives to petroleum-based polymers. In this regard, paper-based products have turned out to be a promising choice; however, their weak resistance to water has limited their application. The use of various additives to enhance paper’s moisture resistance is a common practice. However, considering the growing global agenda for sustainable development, the search for new bio-based paper additives has become increasingly important. This study investigated the potential synergistic impact of the addition of nanofibrillated cellulose (NFC) and chitosan additives (CHIT) to different fiber combinations to improve paper’s properties, in particular, their wet strength. The efficacy of the additive application order was examined and was found to be crucial in achieving the desired outcomes. The results showed that incorporating CHIT after NFC enhanced the paper’s tensile and burst indicators, as well as the paper stretch in the dry state, by 35–70%, 35–55%, and 20–35%, respectively. In addition, the tensile index and stretch in the wet state improved 9–13 times and 2.5–5.5 times over, respectively. The air permeability decreased 2.5–12 times over. These findings demonstrate that the sequential addition of the NFC and CHIT additives yield a greater enhancement of paper’s properties than using each additive separately.

## 1. Introduction

As global concerns regarding environmental sustainability continue to gain momentum, the packaging industry has been pushed to find innovative solutions that not only cater to the functional needs of the packaging but also ensure eco-friendliness [1,2]. In the pursuit of reducing plastic usage, the paper industry has resurged as a cutting-edge alternative [3]. Paper packaging has many advantages, including being biodegradable, renewable, and recyclable [2]. Although paper packaging provides a substitute for plastic, its inherent fragility when in contact with moisture has led manufacturers to incorporate chemical additives to provide wet strength [4,5,6]. These additives, such as synthetic resins, adversely affect the recycling process and the environment in general [7]. Manufacturers therefore need to find novel biodegradable and sustainable additives to enhance the wet strength properties of paper packaging [8,9].

Additives like chitosan and nanofibrillated cellulose (NFC) have emerged as sustainable and abundant raw alternatives that are capable of improving the wet strength of paper packaging without compromising its biodegradability [10,11,12].

Chitosan is a natural polymer that has been used in various applications, including biomedical, agricultural, and industrial [13]. It is derived from chitin, a natural biopolymer that is found in the shells of crustaceans like crabs and shrimp. Chitin is the second most abundant natural polymer in the world after cellulose [11]. Chitosan can be obtained by deacetylating chitin, which removes the acetyl groups from the molecule and leaves behind a positively charged amino group [11,12]. The degree of deacetylation of chitosan can vary, affecting the properties of the molecule, including its solubility, viscosity, and reactivity [13,14]. Chitosan has been shown to improve the wet strength of paper by crosslinking with cellulose fibers [15,16,17,18,19,20,21]. The crosslinking between the chitosan and cellulose fibers is due to the electrostatic interaction between the positively charged amino group of chitosan and the negatively charged hydroxyl groups of cellulose. This interaction can form hydrogen bonds, covalent bonds, and a polyelectric complex, which stabilize the cellulose fibers and prevent them from disintegrating in the presence of water [11,13,15,17,19]. Chitosan can also form a barrier on the surface of paper that protects it from water damage [18,20,22].

NFC is a nanomaterial derived from cellulose fibers. It has a high surface area, high aspect ratio, and high mechanical strength, which makes it an effective reinforcing agent for paper-making. NFC can enhance the mechanical properties of paper by forming a network of nanofibers and cellulose due to hydrogen bonding. NFC fills the space between the fibers, thus increasing the mechanical strength and reducing the porosity and permeability of air and oxygen. The high surface area of NFC allows for a high degree of interaction with the cellulose fibers, resulting in a strong and durable paper product [4,10,23,24].

When combined, chitosan’s high cationic charge density favors interaction with the negatively charged fibers of cellulose-based materials, while NFC’s high surface area and large number of hydroxyl groups provide the ideal platform for nano-scale interfibrillar bridging. The synergistic effect between the two additives can be attributed to the intimate interaction between the cationic functional groups of chitosan and the NFC’s hydroxyl groups, as well as their high surface area, which facilitates strong bonds [25,26,27,28]. Studies on chitosan films with NFC additives have shown that the addition of NFC improved the mechanical and barrier properties of chitosan films, thus providing insight into the potential synergy of chitosan and NFC as additives in improving paper’s properties [29,30,31,32,33,34,35].

A considerable amount of research has focused on the properties of chitosan and NFC as individual additives for paper packaging. The use of chitosan has improved paper’s wet strength, water resistance, and mechanical properties, while NFC has proved to be useful in reducing the porosity of paper [10,11,15,36,37,38,39]. However, more research needs to be conducted on the synergistic effect of these additives, and their optimal concentration required for the enhanced wet strength of paper.

To the best of our knowledge, only the studies by Abdallah et al. [40] and Elgat et al. [39] have addressed the synergy of NFC and chitosan in improving paper’s properties. As an addition to the previous research conducted by Abdallah et al. and Elgat et al., our current study delves into a deeper investigation of ammonium-persulfate-oxidized NFC in tandem with a chitosan solution, and continues our previously described research on improving recycled paper materials through the incorporation of virgin fibers [41]. Our study is unique in the examination of the varied outcomes of the addition order of these agents, demonstrating noteworthy differences, as well as exploring the multifaceted impact of additives on paper containing diverse fiber combinations.

## 2. Materials and Methods

### 2.1. Materials and Reagents

The recovered paper fibers (RFs) of an industrially recycled pulp were supplied by SIA V.L.T. (a producer of molded fiber products, Valmiera, Latvia). The RFs were composed of 60% MIX wastepaper (consisting of newspapers, journals, books, office paper, and packaging paper), 30% waste cardboard, 5% printing house waste, and 5% waste from the egg packaging production process.

The unbleached softwood kraft pulp fibers (KFs) were acquired through the kraft pulping process using a 2 L laboratory digester at a temperature of 170 °C, with a cooking time of 3 h. The process parameters involved an active alkali concentration of 57.4 g/L of NaOH, a sulfidity of 29.8%, and a liquid-to-wood ratio of 4.5 L/kg.

Industrial hemp *Cannabis sativa* (USO-31) was the source of the hemp fibers (HFs). The hemp underwent decortication before being treated in a 2 L laboratory digester with a 4% NaOH solution at 165 °C for 75 min. Subsequently, the fibers were rinsed with tap water for neutralization and refinished using the Blendtec 725 (Orem, UT, USA) at a 1.5% concentration with 179 W for 7 min. Afterward, the fibers were dried at room temperature and kept dry until used. Table 1 summarizes the primary information on the fiber materials used in the study. A more extensive description can be seen in our previously published study [41].

The nanofibrillated cellulose (NFC) was obtained as previously developed using our team’s method [42,43] and is described below.

The chitosan additive (CHIT) was prepared using the method mentioned below.

#### 2.1.1. Preparation of Ammonium-Persulfate-Oxidized NFC

A method for oxidizing bleached hardwood kraft pulp, kindly provided by Metsä Fiber (Metsä, Finland), was employed in this study. The pulp was initially soaked in 1 L of distilled water for 8 h and then disintegrated in a disintegrator for 75,000 revolutions. After removing the excess water, the pulp was repeatedly mixed with 1 L of fresh distilled water in a 2 L glass beaker, covered with foil, and heated to 70 °C in a water bath. A suspension of 100 g ammonium persulfate (Fisher Chemical, Pittsburgh, PA, USA, puriss, ≥98%) was added to the mixture and was stirred continuously at 70 °C for 4 h. The oxidation process was halted by placing the beaker in an ice bath and cooling the mixture to 15 °C. The treated fibers were then washed until neutral and kept at 4 °C.

For the mechanical treatment, a suspension of approximately 1.5% *w*/*w* oxidized fibers was prepared and subjected to sonication using an ultrasonic homogenizer (SONIC-650W, MRC Ltd., Holon, Israel) for 15 min at 90% power, with the settings of 9 s on and 1 s off, to ensure the proper dispersion of the fibers and to prevent clogging the equipment. The resulting suspension was further processed using a microfluidizer (LM20, Microfluidics, Quadro Engineering, Waterloo, ON, Canada) with a 200 µm ceramic chamber H30Z three times, followed by the addition of a 100 µm diamond chamber H10Z and processing at 300–600–900–1500 bar, with three passes at each pressure. Treatment at 2000 bar was then performed for 6 passes, with cooling in the ice bath during the process. A semi-transparent viscous solution was obtained and stored at 4 °C until further use [42,43]. Characterization of ammonium-persulfate-oxidized NFC are summarized in Table 2.

#### 2.1.2. Preparation of the CHIT

10 g food-grade chitosan (Jiangsu Aoxin Biotechnology Co., Ltd., Lianyungang, China), with a deacetylation degree >90% and middle molecular weight, was dissolved in 1 L of 1% acetic acid (Sigma-Aldrich, St. Louis, MO, USA, puriss, ≥99.8%).

### 2.2. Methods

#### 2.2.1. Fiber Handsheet Preparation

Enough fibers to obtain 10 handsheets of 75 g m^−2^ was placed into a glass beaker and soaked in 1–2 L of distilled water for 8 h. The fibers were disintegrated in the disintegrator for 75,000 revolutions (FRANK-PTI, Laakirchen, Austria) before adding NFC and/or 1% chitosan solution in acetic acid to the fiber suspension to achieve the necessary weight ratio based on the dry mass of the fibers. In preparation for the paper stock, the additives were stirred with the suspension at 3500 rpm for 1 min. The fiber sheets were produced using a Rapid-Köthen paper machine (PTI, Laakirchen, Austria) in accordance with ISO 5269-2:2004 [44]. The samples were named based on their composition, including abbreviations of the fibers and mass proportion in percentage, except for NFC and CHIT, which were incorporated as additives into the entire composition. For example, composition MK 50 KF 25 HF 25 + NFC indicates that it contains 50% MK, 25% KF, and 25% HF, with an added 3% NFC based on the dry fiber mass. “NFC + CHIT” indicates that NFC and chitosan were added to the fiber suspension simultaneously. “1. CHIT; 2. NFC” denotes that after the chitosan solution in acetic acid was mixed into the fiber suspension, the NFC was added. “1. NFC; 2. CHIT” implies that the NFC was added initially and mixed into the suspension, followed by the addition of the 1% chitosan solution in acetic acid.

For the samples containing NFC, 3% of NFC based on the dry fiber mass was added throughout all the groups.

In the research, the synergistic effects of the NFC and CHIT were examined by using 2.5% chitosan. This percentage was selected in accordance with our previous research [18,19,20] and provided the baseline for further experiments.

#### 2.2.2. Characterization of the Paper Properties

For the mechanical testing, paper strips of 1 cm width were prepared using a strip cutter (FRANK-PTI, Austria). All the samples were conditioned at 25 °C and a RH of 50% before testing.

The tested samples were subjected to the ISO 1924-1:2008 [45] and ISO 2758:2014 [46] criteria using FRANK-PTI’s vertical tensile tester F81838 and burst tester, correspondingly. Average measurements of the burst index, tensile index (wet and dry state), and stretch (wet and dry state) were acquired from 12 samples. To determine the tensile index and wet stretch, the paper strip was immersed in water for 30 s, and, after removing the excess water, the measurement was immediately performed.

To test the water adsorption, a Cobb tester was used for 60 s in accordance with TAPPI standard method T 441 om-98 [47] with minor modification—the tests were performed only from the wire side of the handsheet. Five measurements on the wire side were conducted on each sample, and the average result was reported.

A Bendtsen tester (Lorentzen & Wettre, Kista, Sweden) was employed for the air permeability tests, followed by the ISO 5636:2019 [48] procedure. The average value from 15 measurements was recorded for the results on the air permeability.

#### 2.2.3. SEM

In order to analyze the fiber, chitosan, and NFC interaction in the paper system using scanning electron microscopy (SEM), a K550X sputter coater (Emitech, Chelmsford, UK) was used to coat them with gold plasma. The examination was conducted using the Tescan Vega TX from Brno, Czech Republic, equipped with software version 2.9.9.21.

#### 2.2.4. Statistical Analysis

The Excel data analysis feature was employed to conduct the statistical analysis. ANOVA (single-factor) was utilized to investigate the correlations among various variables. All the statistical tests were performed with the significance level set at α = 0.05. Consequently, any *p*-value equal to or smaller than 0.05 was deemed statistically significant, resulting in the rejection of the null hypothesis.

## 3. Results and Discussion

The research involved conducting trials on five distinct fiber compositions, both with and without the addition of 3% NFC and 2.5% CHIT. This ratio of chitosan was determined as the most effective from our earlier studies with pure cellulose and wastepaper fibers [18,19,20]. The research delved into understanding how the introduction of NFC and chitosan in varying sequences influenced the characteristics of the fibrous compositions.

### 3.1. Burst Index

The benefits of incorporating HFs or KFs into the fiber composition are clearly illustrated in Figure 1, where a statistically significant improvement in the burst index was observed compared to with pure RFs. The HF content yielded a more pronounced effect than the KF content. The results show that the addition of CNF or CHIT separately to the fiber combinations had an impact on the burst index. While the increase in the burst index for RF100 and RF75HF25 with the addition of NFC or CHIT was not statistically significant, the addition of these additives to RF75KF25, RF50HF50, and RF50HF25KF25 resulted in a statistically significant noteworthy increase of 30–45% within the margin of error. No significant difference was observed between NFC addition and CHIT addition for all the sample types. It is possible that fiber combinations containing more HFs and KFs had more inter-fiber spaces in the paper cast, which were then filled by the NFC particles or CHIT, forming a film between the fibers, and thus significantly increasing the burst index. Other authors have also observed an increase in the burst index with the addition of CHIT or NFC [11,17,39,49,50]. The synergy between the NFC and CHIT enhanced the burst index even more, as demonstrated in Figure 1. Importantly, the sequence in which both components were added was crucial. Sequential addition yielded a considerably higher burst index as opposed to simultaneous addition. Our results statistically significantly revealed burst index improvements according to the sequential addition of NFC and CHIT into the KF-containing samples, resulting in a 70–80% increase. In contrast, the samples containing solely RFs and HFs showed a statistically significant moderate increase of 45–55%. This could be explained by the fact that a paper sheet containing KFs has more inter-fiber spaces.

Elgat et al. reported comparable findings by including into flax fiber molds 4% CHIT, resulting in a 20% increase, and by incorporating a nanocopolymer of CHIT + 3%NFC, thus resulting in a 50% increase. The study conducted by Abdallah et al. revealed mere 2–7% increments in the burst index values. However, it is worth mentioning that their analysis was conducted on virgin pulp, which displays significantly higher absolute values in the burst index. In comparison to our investigation, Abdallah et al. examined chitosan concentrations up to 2% and incorporated NFC at concentrations of 5% and 10% [40].

It is known that when NFC and CHIT are combined, polyelectrolyte complexes are formed [16,33,51]. Initial addition of NFC fills the gaps between fibers [5,7], while the addition of CHIT results in a sturdy film in the inter-fiber space, consequently enhancing the paper’s properties [15,18,39,49].

The SEM pictures in Figure 2 clearly show and confirm the above description of how the chitosan additive forms films in the inter-fiber space and forms film-like bridges between the fibers (b), and how the NFC additive incorporates the chitosan films formed in the inter-fiber space (c) in comparison with paper without additives (a), thus providing an increase in the mechanical properties of the paper.

### 3.2. Tensile Index and Wet Strength

As can be seen from Figure 1 and Figure 3, the effect of the fiber combinations with the NFC and CHIT additives on the changes in the tensile index in the dry state are similar to the burst index. In the fiber combinations containing KFs without additives, no statistically significant increase in the tensile index was observed compared to the pure RFs. At the same time, the presence of HFs in the fiber combination increased the tensile index by 30–40%. The results show that the introduction of the NFC or CHIT additives did not bring statistically significant changes to the tensile index value for the RF100 and RF75HF25 fiber combinations, mirroring the observations of the burst index. However, the results indicated a noteworthy increase in the tensile index values for the RF75KF25, RF50HF50, and RF50KF25HF25 fiber combinations, with a statistically significant increase of 27–44%. Fiber combinations with HFs and KFs may possess larger gaps between the fibers, which can be filled by NFC particles or CHIT to generate a film between the fibers, resulting in a substantial boost in the paper’s tensile index. These data are corroborated by the studies of Khantayanuwong et al. [21] and Bhardwaj et al. [17], who reported comparable findings. Furthermore, Hamzeh et al. [52] noted a substantial enhancement in the tensile index upon the incorporation of chitosan. The results reveal that, akin to the burst index, the inclusion of both NFC and CHIT led to a marked increase in the tensile index values. However, it was noted that the additive inclusion order was more critical to the tensile index in the dry state than to the burst index.

The results show that for the fiber combinations, such as RF100 and RF75HF25, the simultaneous addition of NFC and CHIT resulted in a statistically significant increase in the tensile index. Yet, for other fiber combinations, simultaneous addition did not result in any statistically significant enhancement in the tensile index, whereas the addition of either NFC or CHIT alone. As for the burst index, the fiber combinations incorporating KFs exhibited a statistically significant surge in the tensile index after the gradual incorporation of NFC and CHIT, achieving 70% escalation. The samples that comprised only RFs and HFs showed a moderate statistically significant increase of 35–50% in their tensile index values. One possible explanation could be that fiber compositions containing KFs have a higher number of inter-fiber gaps. The use of a pre-prepared copolymer additive containing CHIT and NFC in Elgat et al.’s [39] study resulted in a significant 30% improvement in the tensile index, similar to the results we observed in our study when both additives were incorporated at the same time. Abdallah et al. [40] study obtained a 7–9% improvement in the tensile index when chitosan was added first, followed by NFC.

The findings presented in Figure 4 demonstrate that fiber combinations containing KFs and HFs without additives exhibited a statistically significant increase in the tensile index in their wet state compared to with RFs. The tensile index values for the KF/HF-containing fiber compositions were 75–85% higher. These findings show that when NFC was added to a fiber composition of RFs, the tensile index in the wet state increased by 125%. The addition of NFC to the compositions containing 75% RFs statistically significantly boosted the wet strength, increasing the tensile index by between 50 and 60%. The addition of NFC to the fiber compositions containing 50% RFs yielded a 70% statistically significant improvement in the tensile index. The data in Figure 4 clearly illustrate that the introduction of chitosan into the fiber mixtures had a remarkable statistically significant impact. The wet strength increased 5.5 to 8 times over in the cases of RF75HF25, RF75KF25, RF50HF50, RF50HF25KF25, and RF100, in the respective order. Similar results were obtained also in our previous studies [18,19,20]. The sequence of additive addition was found to be crucial in achieving the desired results for an enhanced tensile index in the wet state. The most statistically significant improvement was observed when the NFC was added first, followed by the CHIT. Interestingly, when the NFC was added after CHIT’s inclusion into the pure RF fiber composite, the results were inferior to those with CHIT alone. At the same time, the addition of NFC and CHIT together for the samples containing KFs did not statistically significantly change the wet strength compared to the addition of CHIT alone. These findings highlight the importance of carefully considering the order of combining additives into composite materials to optimize their properties.

The results showed an improvement in the tensile index value upon the addition of NFC and CHIT, with a statistically significant synergy observed when the two additives were introduced sequentially. In the case of the RF100 sample, the tensile index value increased 13 times, while other fiber combinations recorded a 9-time increase. These outcomes indicate that the sequential addition of NFC and CHIT can increase the bond strength between the fibers, resulting in a substantial enhancement in paper strength. In particular, the synergistic effect of the additives was evident in the wet strength performance, where the sequential addition of both additives resulted in a tensile index value higher than the sum of the individual effects of each additive. In their research, Abdallah and colleagues [40] introduced chitosan as the first additive into a fiber combination before adding NFC. The rationale was that chitosan can form various bonds with cellulose, including polyelectrolyte complexes, hydrogen, and covalent bonds. NFC, which carries a positive charge, is expected to join the layer of chitosan and form a strong link. However, our findings suggest that the interaction between chitosan and NFC may be more complex. When chitosan is added first, it may bind with the cellulose fibers so tightly that it cannot establish a bond with NFC. On the other hand, when NFC is added first, it can form hydrogen bonds with the cellulose fibers, enabling chitosan to bind with both the NFC and cellulose fibers, and create a film coating. Elgat et al. [39] used NFC and CHIT simultaneously as a pre-prepared copolymer additive. If chitosan and NFC are added simultaneously, they may compete among themselves for binding sites and form weaker links. These findings indicate that the sequence of adding additives to fiber combinations has a significant effect on crosslinking and ultimately affects the quality of the paper produced.

After a thorough analysis of the mechanical outcomes, it has been determined that the optimal results in terms of the absolute values were achieved when using fiber combinations containing 50% RFs and implementing the sequential addition of NFC and then CHIT. Upon assessing the effect of the additives on wet strength and burst, the most significant improvements were witnessed in the RF100 samples that were infused with sequentially added NFC and CHIT. In contrast, under dry conditions, the most considerable surge in the tensile index was observed in the samples containing KFs with the sequential addition of NFC and CHIT.

### 3.3. Stretch

The findings of our study revealed that RF100 is the fourth most flexible fiber combination in the dry state, while the fiber combinations containing HFs demonstrated the least flexibility or stretch. Interestingly, the addition of additives to the RF100 sample did not show any statistically significant increase in the stretch value. In contrast, the inclusion of CNF or NFC into the fiber combinations with HFs statistically increased the stretch value by 20–35%. Similar results have been obtained by other authors [17,52]. The simultaneous or sequential addition of NFC and CHIT did not show a statistically significant increase in the stretch values compared to the samples that included CHIT separately, except for the sample RF75HF25, whose stretch value increased by 45%. The research conducted by Elgat et al. [39] and Abdallah et al. [40] did not involve the evaluation of the stretch changes that occurred when NFC and CHIT were combined.

The results presented in Figure 5 and Figure 6 show the contrast between the elasticity observed in the wet and dry states. With respect to stretchability, the RF100 sample exhibited the lowest value, which was almost half that of the other samples tested. The inclusion of NFC in RF100 enhanced its wet state stretch by 120%, while the other samples exhibited a 15–35% increase, eventually achieving nearly identical stretch absolute levels within the error margins. CHIT, on the other hand, caused a four-fold increase in stretch for RF100 and a 2–2.5-fold increase for the other samples, resulting in comparable absolute stretch values within the error margins. As already demonstrated in the research on the wet strength, the combination of NFC and CHIT required a specific order of addition to yield significant results. If CHI was introduced before NFC or if both were added in tandem, the stretch value in the wet state would not exhibit a notable increase compared to that of CHIT alone. When NFC was introduced first, followed by CHIT after mixing, the resulting combination formed a potent synergy that significantly raised the wet strength. In fact, the cumulative impact of NFC and CHIT was greater than the sum of their individual benefits. The research data showed that the addition of CHIT after NFC incorporation resulted in a 5.5-fold increase in the stretch value of the RF100 sample, and a 2.5–2.8-fold increase across the rest of the samples. The inclusion of NFC and CHIT has led to comparable outcomes in RF100 and the fiber combinations incorporating HFs, with all the results falling within the error margins. However, the fiber blends containing KFs demonstrated slightly lower values. Overall, the stretch values for the samples comprising KFs with the subsequent addition of NFC and CHIT displayed the best absolute values in the dry state. However, the fiber combinations including HFs demonstrated a notable improvement in the dry state. In contrast, in the wet condition, RF100 and the samples containing HFs with the sequential addition of NFC and CHIT exhibited the highest absolute stretch values, yet RF100 emerged as the sample with the most significant increase.

### 3.4. Cobb

Similar to the previous values characterizing the paper’s properties, a decrease in the paper’s water absorbance was also observed in the Cobb test (Figure 7). The addition of NFC statistically significantly reduced the Cobb value by 14% and 16% for the RF100 and RF50HF50 samples, respectively, and by 8% for RF75HF25. For the samples containing KFs, the addition of NFC did not result in a statistically significant increase in the Cobb value.

Accordingly, the addition of chitosan reduced the Cobb water absorption value by 38% for the sample containing only recovery fibers, by 25% for RF75HF25 and RF50KF25HF25, and by 30% other samples. It is interesting to observe that the addition of NFC together with CHIT does not statistically significantly affect the Cobb value within the error limits compared to the Cobb values of adding CHIT alone. It can be assumed, while chitosan has a greater effect on the barrier properties, like decreasing the water absorbance, the synergy of NFC and CHIT greatly increase the wet strength.

### 3.5. Air Permeability

Air permeability, or the lack of it, is an important indicator of packaging material, especially when it comes to food packaging. The study’s findings, as depicted in Figure 8, reveal that the fiber compositions incorporating KFs had inflated air permeability levels, indicating more void spaces between the fibers. These observations lend credence to the previous hypothesis detailed under Section 3.1, indicating that there is a direct correlation between expanded inter-fiber gaps and the ability of NFC and CHIT to occupy that space, thereby enhancing the mechanical traits profoundly. Simultaneously, the combination of the fibers and HFs resulted in limited air permeability.

The addition of HFs to the fiber combinations made it possible to statistically significantly improve the barrier properties of the packaging material. The addition of NFC or CHIT did not statistically significantly affect the air permeability of RF100 within the error margins. This could be attributed to the presence of fillers in the waste fibers that compete with NFC and CHIT. However, when NFC or CHIT was added to RF75KF25, it reduced the air permeability by 25% and 40%, respectively. When NFC or CHIT was added to RF75HF25, the air permeability decreased by 50% and 40%, respectively. Furthermore, the addition of NFC or CHIT to RF50HF50, which already had the lowest air permeability, reduced the air permeability nine and eight times over, respectively. In addition, the air permeability of the sample RF50KF25HF25 decreased three times over with the addition of NFC or CHIT. Bhardwaj et al. [17] obtained a three-fold reduction in the air permeability with chitosan addition. The impact of the order of NFC and CHIT addition was explored once again. While the results in Figure 8 suggest that NFC and CHIT showed comparable and no statistically significant effect when added simultaneously or consecutively with NFC introduced first, it is evident that introduction of NFC after CHIT failed to yield any additional benefits in comparison to CHIT alone. The addition of both additives statistically significantly reduced the air permeability in the case of RF100 and RF75KF25 2.5 and three times over, respectively. Similarly, for RF75HF and RF50KF25HF25, the air permeability decreased five times over, while for the fiber composition RF50HF50, which initially had the lowest air permeability, decreased 12 times.

## 4. Conclusions

This study has upheld the hypothesis that the combination of NFC and CHIT additives would yield higher enhancements to paper’s fiber properties than their individual usage. Specifically, the wet strength and stretch in wet conditions were improved by the synergistic effect of the sequential addition of both additives. Interestingly, the order in which the additives were added was found to be crucial, with the combination of NFC followed by CHIT yielding the highest efficacy. Incorporating chitosan after NFC increased the tensile/burst indicators and stretch in the dry state by 35–70%, 35–55%, and 20–35%, respectively. At the same time, there was a significant improvement in the tensile index and stretch in the wet state ranging from 9–13 times and 2.5–5.5 times over, respectively. There was a notable decrease in air permeability ranging from 2.5 to 12 times over. In conclusion, the fiber combinations with 50% hemp fibers and the sequential addition of NFC and CHIT showed the most desirable mechanical and barrier properties in their absolute values, while the recovered paper fibers showed the most significant improvements.

## Figures and Tables

**Figure 1 polymers-16-00227-f001:**
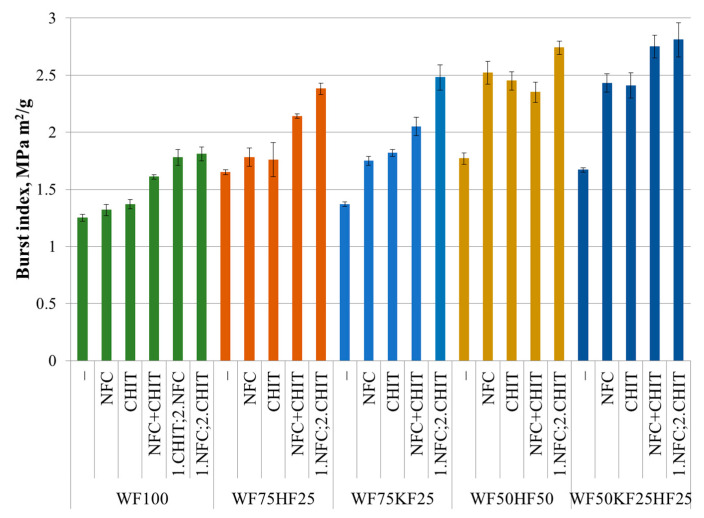
Burst index (dry state) of fiber compositions with and without 3% NFC and 2.5% chitosan additive.

**Figure 2 polymers-16-00227-f002:**
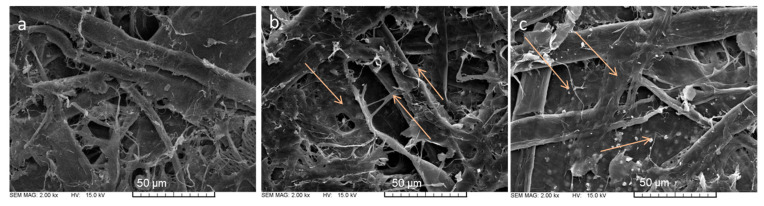
SEM images of paper consisting of 50% recovered fibers and 50% hemp fibers at 2000× magnification (**a**) without additives, (**b**) with 2.5% CHIT addition, (**c**) with 3% NFC and 2.5% CHIT.

**Figure 3 polymers-16-00227-f003:**
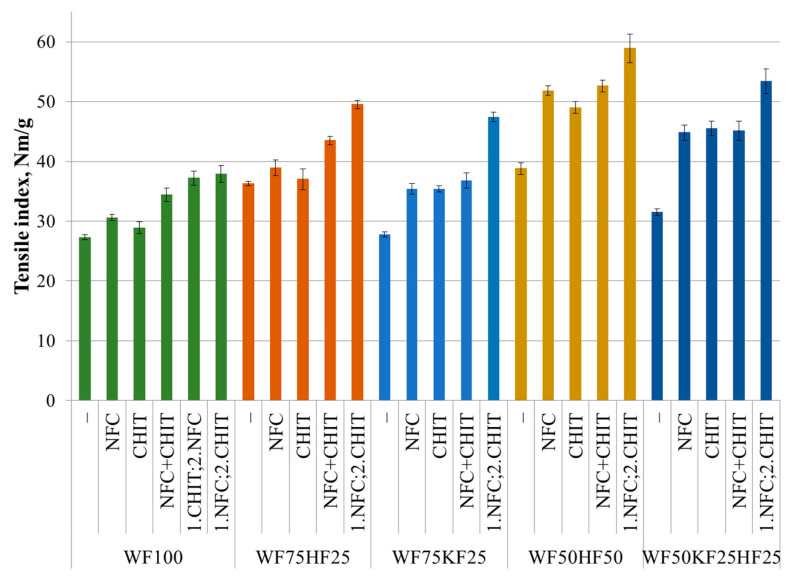
Tensile index (dry state) of fiber compositions with and without 3% of NFC and 2.5% chitosan additive.

**Figure 4 polymers-16-00227-f004:**
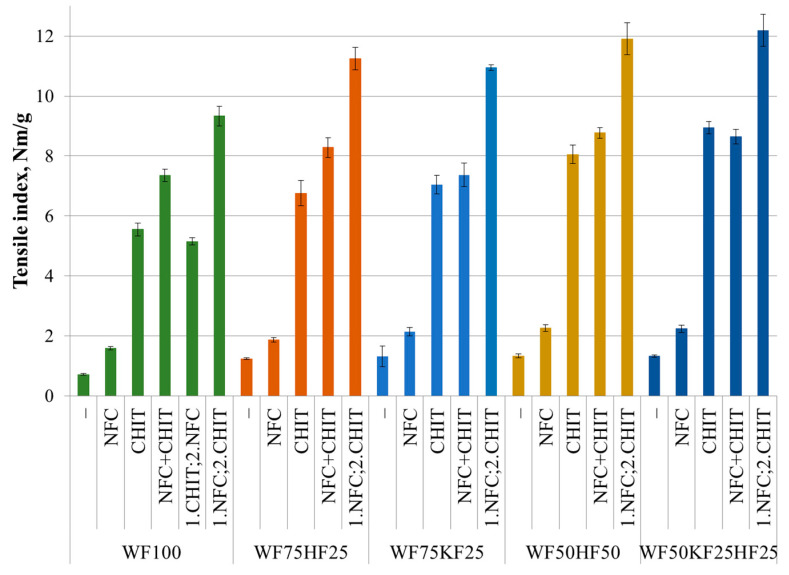
Tensile index (wet state) of fiber compositions with and without 3% of NFC and 2.5% chitosan additive.

**Figure 5 polymers-16-00227-f005:**
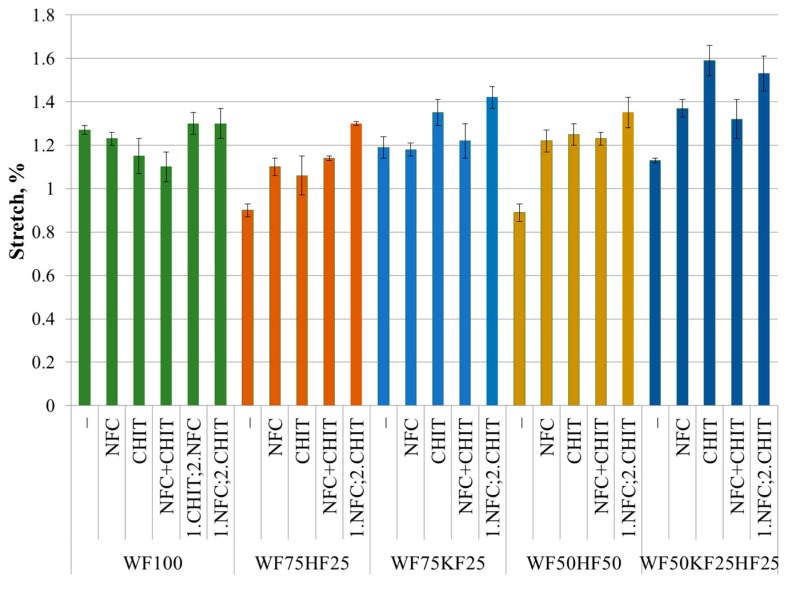
Stretch (dry state) of fiber compositions with and without 3% NFC and 2.5% chitosan additive.

**Figure 6 polymers-16-00227-f006:**
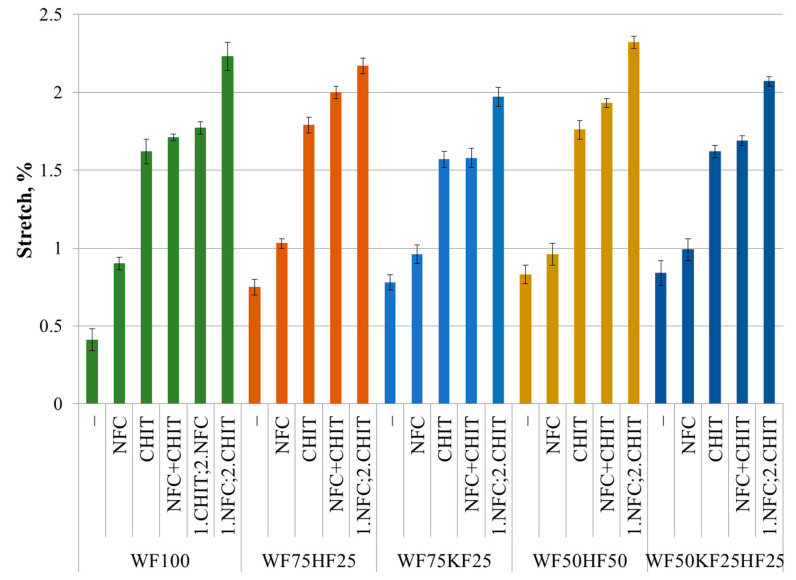
Stretch (wet state) of fiber compositions with and without 3% NFC and 2.5% chitosan additive.

**Figure 7 polymers-16-00227-f007:**
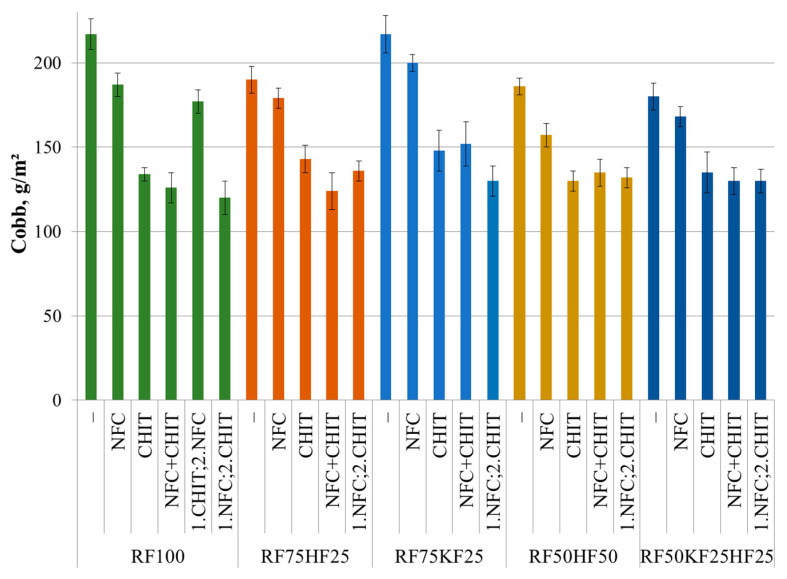
Cobb value of fiber compositions with and without 3% NFC and 2.5% chitosan additive.

**Figure 8 polymers-16-00227-f008:**
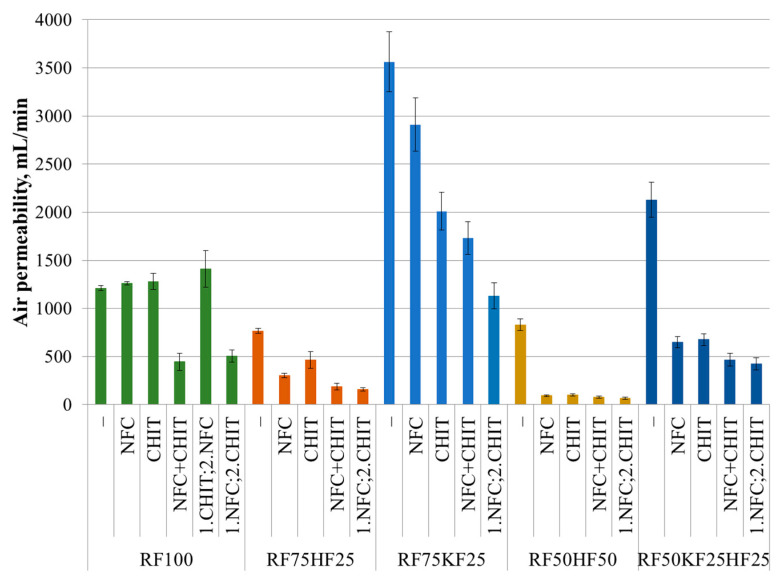
Air permeability of fiber compositions with and without 3% NFC and 2.5% chitosan additive.

**Table 1 polymers-16-00227-t001:** Characterization of the fibers used in the study.

Fibers	Fiber Length, mm	Fiber Wid, µm	Fiber Shape, %	Content of Fines, %
RF	1.19 ± 0.06	25.7 ± 0.4	89.1 ± 0.1	9.7
KF	2.16 ± 0.02	29.6 ± 0.4	90.9 ± 0.1	3.2
HF	0.65 ± 0.01	18.8 ± 0.4	91.8 ± 0.1	7.4

**Table 2 polymers-16-00227-t002:** Characterization of NFC made in previous research [43].

Cationic Demand, µeq/g	WRV, gH_2_O/g	Specific Surface Area, m^2^/g	Degree of Polymerization
333 ± 7	1.7 ± 0.11	154	475 ± 15

## Data Availability

Data will be available upon reasonable request.

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
