# Peer review of "Enhancing Paper Packaging’s Wet Strength Using the Synergy between Chitosan and Nanofibrillated Cellulose Additives"

_polymers, 2024, doi:10.3390/polym16020227_

Round 1
Reviewer 1 Report
Comments and Suggestions for Authors
Abstract is correct.
The introduction is correct with a small reservation that there are no review articles on the impact of NFC and CHIT on paper properties. E.g. article on A comprehensive review of chitosan applications in paper science and technologies by Mostafa Rohi Gal et al.
In the section on research methodology, a description of the symbols used in the abbreviations should be included in the section on the research material. In subsection 2.2.3. regarding Characterization of paper properties, there are no parameters at which the measurements were carried out.
In the part regarding results, there are no results of the authors' own research on the impact of CHTI content other than 2.5%. And in lines 160 and 161, the authors mention that they will conduct such research. The authors only refer to research by other authors. Moreover, the term "nano fibrillated cellulose" is used throughout the article as NFC, while in the captions of Figures 1-6 they use the abbreviation CNF. I know that these abbreviations can be used interchangeably, but for the readability of the article it would be good to use one symbol.
The conclusions do not mention the impact of CHTI at a concentration other than 2.5%.
Author Response
Dear reviewer,
Thank you for your time devoted to reviewing the manuscript.
In response to your suggestions, comments and corrections:
- Abstract is correct.
- The introduction is correct with a small reservation that there are no review articles on the impact of NFC and CHIT on paper properties. E.g. article on A comprehensive review of chitosan applications in paper science and technologies by Mostafa Rohi Gal et al.
The authors have familiarized themselves with a very valuable specific review article and it has already been included in the manuscript as the 11th reference from the moment the manuscript was submitted.
- In the section on research methodology, a description of the symbols used in the abbreviations should be included in the section on the research material.
All abbreviations were included in the materials section and explained in section 2.2.1., but for easier understanding we created a separate subsection - abbreviations.
- In subsection 2.2.3. regarding characterization of paper properties, there are no parameters at which the measurements were carried out.
Thanks for pointing out the flaw. Manuscript 2.3.3. section has been supplemented with sample conditioning parameters before measurements, as well as a description of tensile test methodology for wet state.
- In the part regarding results, there are no results of the authors' own research on the impact of CHTI content other than 2.5%. And in lines 160 and 161, the authors mention that they will conduct such research. The authors only refer to research by other authors.
The content of the results of this publication is one part of a larger research project, within the framework of which research will be continued on the synergistic effect of chitosan and NFC in different concentrations on the physical-mechanical properties of paper. Realizing that including such information in a publication is confusing, we have removed these sentences from the manuscript. The chitosan concentration of 2.5% was chosen based on our own previous research - 18.-20. reference – line 179-180, line 205-206.
- Moreover, the term "nano fibrillated cellulose" is used throughout the article as NFC, while in the captions of Figures 1-6 they use the abbreviation CNF. I know that these abbreviations can be used interchangeably, but for the readability of the article it would be good to use one symbol.
Thank you for your comment. We made corrections under the image captions.
- The conclusions do not mention the impact of CHTI at a concentration other than 2.5%.
Conclusions no longer include information on chitosan at concentrations other than 2.5%. This manuscript is solely on the effect of chitosan at a specific concentration of 2.5% and NFC at a concentration of 3% of fibrous mass on the mechanical properties.

Reviewer 2 Report
Comments and Suggestions for Authors
The paper describes the enhancement of paper packaging wet-strength with chitosan and nanofibrillated cellulose additives. This topic is already well described and does not cover some new insights. For example, the authors describe the fragility of paper packaging when in contact with moisture but they did not conduct the evaluation of the water vapour transmission rate, one of the most important tests. In addition, some other tests regarding the increase of hydrophobicity could have been conducted. Only the Cobb test was performed but in 4 replicants, which does not have a statistical importance.
The interpretation of the results is manly focused on refrencing the other authors and studies. The interpretation of the authors results is only in tehnical sense - resulting in listing and comparison, without scientific explanation.
Another problem is refrencing to other research in bulk citations.
line 79: not amount of research , but number of studies.. Amount refers to mass, while number refers to count.
line 96- why Wase fibers, it is not the right terminology. It is recovered fibers.
line 104- Latin names are usually written in italic
line 190: please add in some table or somehow what is the meaning of WF75KF25, WF50HF50, and WF50HF25KF25. It is confusing.
Comments on the Quality of English Language
style and gramattical errors
Author Response
Dear reviewer
Thanks for your valuable suggestions.
In response to your comments:
- The paper describes the enhancement of paper packaging wet-strength with chitosan and nanofibrillated cellulose additives. This topic is already well described and does not cover some new insights.
It can be agreed that there is a lot of research on the interplay between chitosan and NFC, particularly in terms of how they bind together and enhance paper properties. However, the majority of publications focus on chitosan and NFC films, or chitosan-NFC coatings on paper, which essentially is a film onto the paper surface. While there have been a few studies exploring how paper properties can be enhanced by adding chitosan to the fiber mass, and only two publications discuss the combined addition of chitosan and NFC to the bulk. Additionally, these studies tend to use lower concentrations of chitosan, and the order in which the additives are added has not been evaluated. This research is crucial for paper product companies that cannot afford to change production technology. By adding chitosan and NFC to the fiber mass before casting the paper, these additives can be incorporated without necessitating changes to the production process.
- For example, the authors describe the fragility of paper packaging when in contact with moisture but they did not conduct the evaluation of the water vapour transmission rate, one of the most important tests.
We concur that the permeability of water vapor plays a crucial role in food packaging, particularly for films and specially coated papers. However, it cannot be agreed that water vapor transmission rate is the most important indicator of hydrophobicity. It is an indicator of barrier properties, but not always a hydrophobic material does not allow water vapor to pass through. The material can be 100% water repellent, but can be porous enough to let water vapor through. The size of a water droplet is larger than the size of water vapor molecules. The hydrophobicity on the surface will be much better described by the contact angle.
Our investigation primarily centers on evaluating the mechanical strength of paper in wet conditions. This attribute is of utmost importance in packaging materials like paper shopping bags or molded egg boxes. While paper bags and egg boxes may not possess substantial barrier properties, their high wet strength becomes critical in situations such as rainfall or the transportation of egg cartons in refrigerators leading to condensation formation.
In future studies, it is planned to analyze the effect of chitosan and NFC on the barrier properties of paper such as air permeability, water vapor permeability, grease resistance.
- In addition, some other tests regarding the increase of hydrophobicity could have been conducted. Only the Cobb test was performed but in 4 replicants, which does not have a statistical importance.
The study includes Cobb tests, wet strength and wet stretch, which is the main focus of our research (when the paper gets wet). Barrier properties for food packaging are not evaluated in our study. Cobb test results show measurements that do not exceed 10% of the measurement value, which is considered normal in the paper industry.
- The interpretation of the results is mainly focused on refrencing the other authors and studies. The interpretation of the authors results is only in tehnical sense - resulting in listing and comparison, without scientific explanation.
The obtained results are compared with the studies of other authors, which is a normal practice in the preparation of scientific articles. Likewise, the manuscript scientifically explained the binding of chitosan and NFC, the formation of polyelectrolytes, why the order of adding NFC and chitosan matters, and why NFC should be added first. However, to further illustrate the increase in mechanical strength, we included SEM images of the paper samples in the manuscript.
- Another problem is refrencing to other research in bulk citations.
The authors of the manuscript did not citate the works of other authors, but rather formed their thoughts in the introduction based on the works of other authors, which is why several sentences have been created in a row, with several references included at the end.
- line 79: not amount of research , but number of studies.. Amount refers to mass, while number refers to count.
Thanks you, corrections made.
- line 96- why Wase fibers, it is not the right terminology. It is recovered fibers.
Thank you. Corrections made to Recovered fibers (RF)
- line 104- Latin names are usually written in italic
Thanks you, corrections made.
- line 190: please add in some table or somehow what is the meaning of WF75KF25, WF50HF50, and WF50HF25KF25. It is confusing.
Thank you. For easier understanding we created a separate subsection - abbreviations.

Reviewer 3 Report
Comments and Suggestions for Authors
Dear Authors
The demand for eco-friendly packaging materials has urged researchers to find an alternative to petroleum-based polymers. In this regard, paper-based products have become a promising choice; however, their water weakness has limited their use in some applications. The use of various additives to enhance the moisture resistance of paper has been a common practice. However, in light of the growing global concern for sustainable development, the search for new bio-based paper additives has become increasingly important. This study investigated the potential synergistic impact of adding NFC and chitosan additives to different fiber combinations to improve paper properties, particularly concerning wet strength. The efficacy of the order of additive application was examined and found to be crucial to achieving desired results. Results showed that incorporating chitosan after NFC enhanced tensile and burst indices, as well
as stretch in dry state, by 35-70%, 35-55%, and 20-35%, respectively; in addition, tensile index and stretch in wet state improved 9-13 times and 2.5-5.5 times, respectively. Air permeability decreased 2.5-12 times. The study's findings demonstrated that adding additives yielded a more significant enhancement of paper properties than using each additive independently.
The following comments may help increase the clarity of the results and help the readers to understand the presented results better.
General comments
1- Water vapor permeability is the essential characteristic of the packaging materials. Why did the authors dismiss such characterization?
2- The antibacterial activity is significant in the packaging materials. Why did the authors dismiss such a study?
3- The thickness of the packaging materials is a determined factor. The authors dismissed such an investigation; why?
4- The authors mentioned using ammonium persulfate to oxidize the NFC used in this study. What was the degree of oxidation they obtained? Have the authors studied the effect of oxidization degree on the properties of the developed packaging films?
Specific comments
In the Abstract, the authors need to mention the names of abbreviations for the first time appearance in the text—for example, NFC.
In conclusion, the study is exciting and addresses a missing point. However,
A major revision is required before reconsidering the manuscript for publication.
Greetings
Comments on the Quality of English Language
A minor revision is needed.
Author Response
Dear reviewer
Thank you for taking the time to review the manuscript and for your very helpful comments.
In response to your main comments:
- Water vapor permeability is the essential characteristic of the packaging materials. Why did the authors dismiss such characterization?
We concur that the permeability of water vapor plays a crucial role in food packaging, particularly for films and specially coated papers. Our investigation primarily centers on evaluating the mechanical strength of paper in wet conditions. This attribute is of utmost importance in packaging materials like paper shopping bags or molded egg boxes. While paper bags and egg boxes may not possess substantial barrier properties, their high wet strength becomes critical in situations such as rainfall or the transportation of egg cartons in refrigerators leading to condensation formation.
In future studies, it is planned to analyze the effect of chitosan and NFC on the barrier properties of paper such as air permeability, water vapor permeability, grease resistance, contact angle.
- The antibacterial activity is significant in the packaging materials. Why did the authors dismiss such a study?
The answer is similar to the previous question, currently our research is more focused on improving the mechanical strength of paper in a particularly wet condition. In addition, in view of the positive attributes of the paper, future studies will also conduct antibacterial tests to validate its safety for use in food packaging.
- The thickness of the packaging materials is a determined factor. The authors dismissed such an investigation; why?
The thickness of the paper samples is measured and further included in the calculation of the mechanical properties according to the standard.
- The authors mentioned using ammonium persulfate to oxidize the NFC used in this study. What was the degree of oxidation they obtained? Have the authors studied the effect of oxidization degree on the properties of the developed packaging films?
Thank you for your comment. In the specific study, the authors have not evaluated the degree of oxidation of NFC, but the properties of the developed NFC have been extensively analyzed in a previous study (reference in the manuscript 42):
Filipova, I.; Serra, F.; Tarrés, Q.; Mutjé, P.; Delgado-Aguilar, M. Oxidative Treatments for Cellulose Nanofibers Production: A Comparative Study between TEMPO-Mediated and Ammonium Persulfate Oxidation. Cellulose 2020, 27, 10671–10688, doi:10.1007/s10570-020-03089-7.
We included the table describing NFC in the methodological section of the manuscript - subsection 2.2.1.
In our opinion, given that the study analyzes chitosan and NFC additives in the bulk of fibers and not as coatings or films,
We have not conducted studies on the effect of NFC on paper properties depending on the degree of oxidation. The study uses NFC with the most optimal oxidation state obtained in previous studies (42). In our opinion, considering that the study analyzes the addition of chitosan and NFC in the fiber mass and not as a coating or films, the degree of oxidation of nanocellulose could have less effect than it would in the case of films or coatings. Nevertheless, thanks for the comment and we will consider doing a study with different oxidation states of NFC in the future.
Specific comments
- In the Abstract, the authors need to mention the names of abbreviations for the first time appearance in the text—for example, NFC.
Thank you, we included the abbreviation NFC and an explanation in the abstract.
- In conclusion, the study is exciting and addresses a missing point. However, major revision is required before reconsidering the manuscript for publication.
Thank you, thanks to your and other reviewer's recommendations we have made a major revision.
Greetings

Round 2
Reviewer 2 Report
Comments and Suggestions for Authors
Section 2.1. Abbreviations should be in one table for better visability and estetics
you have written that this study will have an positive influence on development of in packaging materials like paper shopping bags or molded egg boxes, since their high wet strength becomes critical in situations such as rainfall or the transportation of egg cartons in refrigerators leading to condensation formation. This is maybe true for the paper bags, but I did not have any bad experice with the egg boxes since the formation of those boxes is very special.
Cobb test is not enough when you are describing your results. Why haven't you conducted some other test to confirm the fromation of hydrophobic properties?
Author Response
Dear reviewer,
Thanks for the suggestions and corrections to improve the publication.
Section 2.1. Abbreviations should be in one table for better visability and estetics.
Thanks for the suggestion. Abbreviations are included in Table 1.
You have written that this study will have an positive influence on development of in packaging materials like paper shopping bags or molded egg boxes, since their high wet strength becomes critical in situations such as rainfall or the transportation of egg cartons in refrigerators leading to condensation formation. This is maybe true for the paper bags, but I did not have any bad experice with the egg boxes since the formation of those boxes is very special.
Our collaborative research project with Egg Box Manufacturing Company Ltd. VLT has focused on addressing the prevalent issue of condensation in truck refrigerators. This issue has had a significant impact on major egg producers, compelling them to switch to plastic boxes. However, with the restriction on the use of plastic in food packaging, these large egg companies are now seeking alternatives to paper packaging boxes that offer wet strenght during transportation. Although the egg cartons do not typically break in users' refrigerators, they face considerable pressure during transportation, where they need to withstand stacking heights of up to 2 meters.
Cobb test is not enough when you are describing your results. Why haven't you conducted some other test to confirm the formation of hydrophobic properties?
The Cobb and Air permeability test is a preliminary measurement of barrier properties to evaluate the further direction of research on food packaging. The main objective of this manuscript is to improve the mechanical strength, especially in the wet condition. We do not claim that the material we developed is hydrophobic, but the wet strenght has increased 9-13 times
Reviewer 3 Report
Comments and Suggestions for Authors
Dear Authors
Thanks for your responding to the raised comments in a satisfactory way.
I can recommend the manuscript for publication.
Greetings
Comments on the Quality of English LanguageA minor revision is needed.
Author Response
Thank you very much!
Best regards
Round 3
Reviewer 2 Report
Comments and Suggestions for Authors
If you didn't want to claim that the material you developed is hydrophobic, why did you describe your results of Cobb test as hydrophobic.
Author Response
Dear Reviewer,
We have changed "hydrophobicity" to "decrease in water absorbance", to reduce misunderstandings.
Happy New Year!
Best regards